# Urinary Long Non-Coding RNA Levels as Biomarkers of Lupus Nephritis

**DOI:** 10.3390/ijms241411813

**Published:** 2023-07-22

**Authors:** Cheuk-Chun Szeto, Ho So, Peter Yam-Kau Poon, Cathy Choi-Wan Luk, Jack Kit-Chung Ng, Winston Wing-Shing Fung, Gordon Chun-Kau Chan, Kai-Ming Chow, Fernand Mac-Moune Lai, Lai-Shan Tam

**Affiliations:** 1Department of Medicine & Therapeutics, Prince of Wales Hospital, The Chinese University of Hong Kong, Shatin, Hong Kong, China; 2Li Ka Shing Institute of Health Sciences (LiHS), Hong Kong, China; 3Department of Anatomical and Cellular Pathology, Faculty of Medicine, The Chinese University of Hong Kong, Shatin, Hong Kong, China

**Keywords:** biomarker, SLE, glomerulonephritis

## Abstract

Background: Emerging evidence suggests that long non-coding RNA (lncRNA) plays important roles in the regulation of gene expression. We determine the role of using urinary lncRNA as a non-invasive biomarker for lupus nephritis. Method: We studied three cohorts of lupus nephritis patients (31, 78, and 12 patients, respectively) and controls (6, 7, and 24 subjects, respectively). The urinary sediment levels of specific lncRNA targets were studied using real-time quantitative polymerase chain reactions. Results: The severity of proteinuria inversely correlated with urinary maternally expressed gene 3 (MEG3) (r = −0.423, *p* = 0.018) and ANRIL levels (r = −0.483, *p* = 0.008). Urinary MEG3 level also inversely correlated with the SLEDAI score (r = −0.383, *p* = 0.034). Urinary cancer susceptibility candidate 2 (CASC2) levels were significantly different between histological classes of nephritis (*p* = 0.026) and patients with pure class V nephritis probably had the highest levels, while urinary metastasis-associated lung carcinoma transcript 1 (MALAT1) level significantly correlated with the histological activity index (r = −0.321, *p* = 0.004). Urinary taurine-upregulated gene 1 (TUG1) level was significantly lower in pure class V lupus nephritis than primary membranous nephropathy (*p* = 0.003) and minimal change nephropathy (*p* = 0.04), and urinary TUG1 level correlated with eGFR in class V lupus nephritis (r = 0.706, *p* = 0.01). Conclusions: We identified certain urinary lncRNA targets that may help the identification of lupus nephritis and predict the histological class of nephritis. Our findings indicate that urinary lncRNA levels may be developed as biomarkers for lupus nephritis.

## 1. Introduction

Lupus nephritis is one of the most frequent and serious manifestations of systemic lupus erythematosus (SLE), accounting for substantial morbidity and mortality in lupus patients [1,2]. Clinical management of lupus nephritis remains a great challenge because of its heterogeneous classification and unpredictable course. However, clinical outcomes of renal involvement can be markedly improved by early diagnosis, close monitoring, and prompt treatment [3]. Kidney biopsy is usually regarded as the gold standard for diagnosis and histological classification of lupus nephritis. Nonetheless, sampling bias is always a possibility, and serial biopsy is often difficult. Urinary biomarkers are therefore an attractive candidate to reflect the activity of lupus nephritis [4,5].

Emerging evidence shows that non-coding RNAs may play a critical role in the regulation of immunological reaction and the development of kidney diseases, and their urinary levels may serve as biomarkers of kidney diseases [6,7]. In addition to microRNA, which has been extensively studied in recent years [7,8], long non-coding RNAs (lncRNAs), defined as RNAs over 200 nucleotides in length that do not encode any protein, are found to be important in controlling gene expression [9]. For example, by using high-throughput RNA sequencing technology to identify Smad3-dependent lncRNAs related to renal inflammation and fibrosis, Zhou et al. [10] found a functional link between progressive kidney injury and two Smad3-dependent lncRNAs. Accumulating evidence has identified a number of lncRNAs that may contribute to the pathogenesis of SLE and lupus nephritis and thus exhibit the potential to serve as clinical biomarkers [11,12,13,14,15,16]. Specifically, the lncRNA MEG3 regulates the interplay between T lymphocyte subsets in SLE through the modulation of transcription factors RORγt and FOXP3 [11]; MGC is involved in the presentation of endogenous antigens [12]; ANRIL plays a pivotal role in the development of inflammatory response in SLE [13]; MALAT1 is the key regulatory factor in the pathogenesis of SLE through regulation of SIRT1 signaling [14]; CASC2 is involved in the regulation of mesangial cell proliferation and the accumulation of extracellular matrices [15]; and TUG1 contributes to the protection of NF-κB inhibition after kidney injury in murine models of SLE [16]. In the present study, we determine the role of these urinary lncRNAs as a non-invasive biomarker for lupus nephritis, specifically aiming to explore potential lncRNA targets for the diagnosis of lupus nephritis, differentiate specific histological classes of nephritis, and detect class V lupus nephritis amongst patients presenting with nephrotic syndrome. Three separate groups of patients with lupus nephritis, with appropriate controls, were recruited to answer these questions.

## 2. Results

### 2.1. Urinary lncRNA for Identification of Lupus Nephritis

In the first study, urinary levels of the lncRNA targets MEG3, ANRIL, and lnc-MGC were tested for the identification of lupus nephritis. We recruited 31 patients with lupus nephritis and six healthy controls. Their baseline characteristics are summarized in Table 1. In essence, urinary ANRIL level was significantly reduced in lupus nephritis compared to healthy controls (0.11 [IQR 0.03–0.59] vs. 0.28 [IQR 0.14–82.54] copies, Mann–Whitney U test, *p* = 0.014) (Figure 1). Urinary lnc-MGC level was marginally elevated in lupus nephritis compared to the control group (1.37 [IQR 0.65–57.37] vs. 0.63 [IQR 0.25–1.40] copies, *p* = 0.053), while urinary MEG3 levels were similar (0.52 [IQR 0.13–2.01] vs. 1.05 [0.25–5.03] copies, *p* = 0.3). With ROC curve analysis, however, the AUC of MEG3, ANRIL, and lnc-MGC for the identification of lupus nephritis was only 0.559 (*p* = 0.7), 0.710 (*p* = 0.14), and 0.597 (*p* = 0.5), respectively.

We further explored the relation between urinary levels of the lncRNA targets MEG3, ANRIL, and lnc-MGC and the severity of lupus nephritis. In essence, the severity of proteinuria inversely correlated with urinary MEG3 (Spearman’s r = −0.423, *p* = 0.018) and ANRIL levels (r = −0.483, *p* = 0.008), while baseline eGFR correlated with urinary lnc-MGC level (r = 0.465, *p* = 0.011) (Figure 2). Urinary MEG3 level also inversely correlated with the SLEDAI score (r = −0.383, *p* = 0.034) and the histological activity index (r = −0.311, *p* = 0.089), although the latter did not reach statistical significance. Urinary ANRIL or lnc-MGC levels did not correlate with the SLEDAI score or histological activity index. None of the urinary lncRNA target levels correlated with the histological chronicity index or the severity of tubulointerstitial fibrosis as measured by the morphometric study (Appendix A).

### 2.2. Urinary lncRNA and Histological Classes of Lupus Nephritis

In the second study, urinary levels of the lncRNA targets MALAT and CASC2 were quantified in 78 lupus nephritis patients with various histological classes, as well as seven healthy controls. Their baseline characteristics are summarized in Table 2. As a whole group, urinary MALAT1 and CASC2 levels were significantly increased in lupus nephritis compared to healthy controls (31.3 [IQR 21.5–61.9] vs. 9.3 [IQR 9.2–11.5] copies and 18.4 [IQR12.2–156.0] vs. 9.9 [IQR 9.7–12.1] copies, respectively; *p* < 0.0001 for both). With ROC curve analysis, the AUC of MALAT and CASC2 for the identification of lupus nephritis was 0.985 and 0.905, respectively (*p* < 0.0001 for both). For MALAT, a cut-off value of 12 provides a sensitivity of 96.2% and a specificity of 100% for diagnosis. For CASC2, a cut-off value of 13 provides a sensitivity of 70.5% and a specificity of 100% for diagnosis.

When the lupus nephritis group was further analyzed, it was found that there was a significant difference in urinary CASC2 levels between histological classes of nephritis (Kruskal–Wallis test, *p* = 0.026), and patients with pure class V nephritis probably had the highest levels (Figure 3). There appeared to be a difference in urinary MALAT1 levels between histological classes of nephritis, but post hoc subgroup analysis did not reach statistical significance after correlation for multiple comparisons. There was a moderate but significant correlation between urinary MALAT1 level and the histological activity index (r = −0.321, *p* = 0.004), though not with baseline eGFR, proteinuria, SLEDAI score, or histological activity or chronicity indices (Appendix A). Urinary CASC2 level did not correlate with any parameters of lupus activity (Appendix A).

After 6 months of immunosuppressive therapy, 34 patients had complete response, 33 partial response, and 11 were refractory to treatment. Neither urinary MALAT1 level nor urinary CASC2 level were associated with treatment response.

### 2.3. Urinary lncRNA and Class V Lupus Nephritis

In the third study, urinary levels of lncRNA TUG1 were compared between 36 patients with pure class V lupus nephritis, primary membranous nephropathy, or minimal change nephropathy. Their baseline characteristics are summarized in Table 3.

Urinary TUG1 levels were significantly different between the three groups (Figure 4). Post hoc analysis showed that urinary TUG1 levels in patients with pure class V lupus nephritis were significantly lower than those with primary membranous nephropathy (*p* = 0.003) and minimal change nephropathy (*p* = 0.04), while there was no significant difference in urinary TUG1 levels between the latter two groups. With ROC curve analysis, the AUC of TUG1 for the identification of class V lupus nephritis was 0.795 (*p* = 0.004). At the cut-off value of 150, TUG1 level had a sensitivity of 79.2% and a specificity of 75.0% for diagnosis. Moreover, urinary TUG1 level significantly correlated with eGFR in patients with pure class V lupus nephritis (r = 0.706, *p* = 0.01) and those with primary membranous nephropathy (r = 0.771, *p* = 0.001), though it did not correlate with minimal change nephropathy (Figure 5). Urinary TUG1 level did not correlate with the severity of proteinuria or the degree of tubulointerstitial scarring in the morphometric study, and this level in the lupus group also did not correlate with SLEDAI score or histological activity or chronicity indices (Appendix A). Because of the small number of patients, the relation between urinary TUG1 level and treatment response was not analyzed.

## 3. Discussion

In this study, we found that urinary CASC2 levels were significantly different between different histological classes of lupus nephritis, and pure class V nephritis had the highest levels. Urinary TUG1 level was significantly lower in class V lupus nephritis compared to primary membranous nephropathy or minimal change nephropathy.

lncRNA is a relatively new field in nucleic acid research, and urinary lncRNA levels have not been well studied. Nonetheless, circumstantial evidence suggests that lncRNAs play an important role in the pathogenesis of SLE and other immunological diseases, and their urinary levels may be valuable markers of kidney diseases. In this study, we asked three questions that are related to the use of urinary lncRNA levels as biomarkers: (1) Can it differentiate patients with lupus nephritis and healthy controls? (2) Can it differentiate patients with lupus nephritis with different histological classes? (3) Can it differentiate class V lupus nephritis from other causes of nephrotic syndrome?

In this study, the choice of lncRNA targets for study was based on circumstantial evidence from previous studies. In the first part, we chose MEG3, ANRIL, and MGC as the lncRNA targets. The lncRNA ANRIL was first reported to contribute to the pathogenesis of diabetic kidney disease via regulation of the cyclin-dependent kinase inhibitor (CKDN) pathway [17]. In this regard, CKDN is involved in the pathogenesis of SLE [18], and ANRIL was recently found to play a pivotal role in the development of the inflammatory response in SLE [13]. On the other hand, the lncRNA MGC contributes to the development of renal fibrosis in early diabetic nephropathy via the regulation of ER Degradation-Enhancing Alpha-Mannosidase-Like Protein 3 (EDEM3) expression [19], a molecule that is involved in the presentation of endogenous antigens [12]. The lncRNA MEG3 modulates microvascular dysfunction via activation of the PI3K/Akt pathway [20]. The imprinted DLK1-MEG3 gene region on chromosome 14q32.2 alters susceptibility to type 1 diabetes and SLE [21]. MEG3 also regulates the interplay between Th17 and Treg cells in SLE through the transcription factors RORγt and FOXP3, which regulate the downstream cytokine network including TGF-β, IL-10, IL-17, and IL-23 [11]. In spite of these theoretical reasons, our results indicated that the differences in urinary MEG3, ANRIL, and MGC levels between lupus nephritis and healthy controls were modest, and no reliable cut-off value could be determined for diagnostic use.

In the second part, we chose MALAT1 and CASC2 as the lncRNA target. Previous studies showed that CASC2 overexpression inhibited the apoptosis of podocyte cells and reduced phosphorylation levels of JNK1 [22]. CASC2 in serum and renal tissue was specifically downregulated in patients with type 2 diabetes with podocyte injury [23], and CASC2 upregulation suppressed proliferation, the accumulation of extracellular matrices, and oxidative stress in mesangial cells through the miR-133b/FOXP1 regulatory axis [15]. On the other hand, MALAT1 is the key regulatory factor in the pathogenesis of SLE due to it exerting detrimental effects through the regulation of SIRT1 signaling [14]. The MALAT1 rs4102217 polymorphism is associated with susceptibility to SLE in humans [24]. The characteristics of CASC2 and MALAT1 suggest that they may affect the phenotype (i.e., histological class) of lupus nephritis.

In the third part, we chose TUG1 as the lncRNA target. Previous studies showed that TUG1 contributed to the protection of NF-κB inhibition after kidney injury in murine models of SLE [16], and TUG1 alleviated LPS-induced mesangial cell injury through regulation of the miR-153-3p/Bcl-2 axis in lupus nephritis [25]. TUG1 is a known regulator of podocyte health [26], and it regulates blood–tumor barrier permeability by targeting miR-144 [27]. These results suggest that TUG1 may be involved in the generation of glomerular permeability barrier damage as well as the regulation of immune complex deposition, which probably explains its correlation with eGFR for lupus nephritis but not MCN.

## 4. Materials and Methods

### 4.1. Overall Study Design

All study procedures were performed in compliance with the Declaration of Helsinki and were approved by the Joint Chinese University of Hong Kong and New Territories East Cluster Clinical Research Ethics Committee (CUHK-NTEC CREC) (approval numbers CRE-2016.480 and CRE-2017.368). This series of work consists of three separate studies. In each study, we recruited patients with active lupus nephritis requiring kidney biopsy. We excluded patients with concurrent infections or chronic hepatitis. Healthy kidney donors or patients with minimal change nephropathy or primary membranous nephropathy were recruited as controls for individual studies. After informed consent was obtained, a whole-stream early morning urine specimen was collected on the date of kidney biopsy for total RNA extraction and expression study. The result of renal function, as represented by the estimated glomerular filtration rate (eGFR), proteinuria, and serological markers of lupus activity, was recorded before kidney biopsy. The eGFR was calculated using the Chronic Kidney Disease Epidemiology Collaboration (CKD-EPI) equation [28].

### 4.2. Preparation of RNA

The methods of urinary sediment isolation and RNA extraction have been described previously [29]. Briefly, urine samples were centrifuged at 3500× *g* for 30 min at 4 °C. Total RNA in urine sediments was extracted using the MirVana™ miRNA isolation kit (Ambion, Inc., Austin, TX, USA) following the manufacturer’s instructions. All samples were pre-treated with Deoxyribonuclease I (Invitrogen, Life Technologies, Waltham, MA, USA) and then stored at −80 °C. For each reaction, approximately 0.5 µg of RNA was reverse transcribed with Superscript II RNase H-Reverse Transcriptase (Invitrogen).

### 4.3. Quantification of lncRNA

We performed real-time quantitative polymerase chain reaction (RT-QPCR) using the ABI Prism 7700 Sequence Detector System (Applied Biosystems, Foster City, CA, USA) to determine the lncRNA level. Based on previous studies [11,12,13,14,15,16,17,19,20,21,22,23,24,25,30], we quantified the lncRNA targets maternally expressed gene 3 (MEG3), antisense non-coding RNA in the INK4 locus (ANRIL), long non-coding mega-cluster (lnc-MGC), metastasis-associated lung carcinoma transcript 1 (MALAT1), cancer susceptibility candidate 2 (CASC2), and taurine-upregulated gene 1 (TUG1). All primer and probe sequences were custom-designed (Applied Biosystems). Small RNA U6 (Applied Biosystems) was used as a house-keeping gene to normalize the lncRNA level [31]. Results were analyzed with Sequence Detection Software version 2.0 (Applied Biosystems) using the ΔΔCT method for relative quantitation, and results were expressed as copy number per 1000 copies of the housekeeping gene.

### 4.4. Histological Study

Kidney pathology was classified according to the revised International Society of Nephrology/Renal Pathology Society system [32]. The histological activity and chronicity indices were also scored by standard means [33]. Briefly, activity index is the sum of the semi-quantitative scores of six lesions and includes hypercellularity, leucocyte infiltration, subendothelial hyaline deposits, interstitial inflammation, necrosis, and cellular crescents. Each lesion is scored from 0 to 3 and the last two items are scored twice. The maximum value of the activity index is 24 points. Chronicity index is the sum of 4 semi-quantitative scores comprising glomerular sclerosis, fibrous crescents, tubular atrophy, and interstitial fibrosis. Each lesion is scored from 0 to 3 and the maximum value is 12 points.

### 4.5. Morphometric Study

Jones’ silver staining was performed on 5 µm thick sections of the renal biopsy specimens of each patient. As previously described by others [34], we used a computerized image analysis method to semi-quantify nephrosclerosis. Briefly, a Leica Twin Pro image analysis system (Leica Microsystems, Wetzlar, Germany) was connected to a Leica DC500 digital camera on a Leica DMRXA2 microscope working with a 40× objective (final calibration: 0.258 mm/pixel), which was connected to a microcomputer for storage of the morphometric measurements so that image analysis could be performed using image analysis software (MetaMorph 4.0; Universal Imaging Corporation TM, Downingtown, PA, USA). A total of 10 glomeruli and 10 randomly selected areas were assessed in each patient and the average percentage of scarred glomerular and tubulointerstitial areas, as represented by the percentage of the area with positive staining, was computed for each patient.

### 4.6. Follow-Up and Treatment Response

All patients with lupus nephritis received standard immunosuppressive therapy as decided by individual clinicians. In the second part, patients were followed for 6 months, and we used the Kidney Disease Improving Global Outcomes (KDIGO) guideline for the definition of clinical response [35]. In essence, complete response is defined as a return of serum creatinine to the previous baseline in combination with a decline in proteinuria to below 0.5 g/day after treatment for 6 months. Partial response is defined as stabilization (±25%) or improvement of serum creatinine, but not to normal, plus a ≥50% decrease in proteinuria. Refractory disease is defined as the failure to achieve at least partial response 6 months after treatment [35].

### 4.7. Statistical Analysis

Statistical analysis was performed using SPSS for Windows software version 24.0 (SPSS Inc., Chicago, IL, USA). All the results are presented as mean ± SD unless otherwise specified. Since the data of lncRNA levels were highly skewed, nonparametric statistical methods were used, including the Kruskal–Wallis H test and the Mann–Whitney U test to compare gene expression levels between groups and Spearman’s rank-order correlations to test associations between lncRNA levels and clinical parameters. The diagnostic value of lncRNA levels was further explored through receiver operation characteristics (ROC) curves and computation of the area under the curve (AUC). A P value of below 0.05 was considered statistically significant. All probabilities were two-tailed.

## 5. Conclusions

Although this study provides some novel data, the development of lncRNA as a biomarker is in its infancy stage and many basic facts remain unknown. For example, urinary lncRNA levels in the normal population have not been defined, and our results indicate that the levels vary from almost undetectable (e.g., MGC, MALAT1, and CASC2) to a substantial amount (e.g., MEG3 and ANRIL). Further validation studies and experiments for delineating the underlying mechanism are necessary. Our work must be regarded as preliminary as there are several major limitations. First, the sample size was small in each individual group, and we only measured urinary lncRNA at the baseline. It would certainly be important to perform serial measurement in order to determine whether urinary lncRNA levels change in response to immunosuppressive therapy, which we did not perform because of limitations in our original study design. We also did not have data related to circulating lncRNA or other cytokine levels for analysis. In addition, our study did not ascertain the cellular origin of lncRNA in urinary sediments. Based on our previous studies, infiltrating mononuclear cells and renal tubular cells are the major origin of messenger RNA, while podocytes represent a non-negligible minority [36,37], though their contribution to lncRNA needs further clarification. Furthermore, since the data in our study came from three different cohorts, there could be the chance of a batch effect. Nonetheless, we did not attempt to remove or rectify the batch effect and instead treated these groups as three separate studies as there was no overlap in the lncRNA targets tested between the cohorts. Finally, our choice of lncRNA target for study was empirical and based on the available literature. Expression profiling of lncRNA would be the ideal approach to identifying potential lncRNA targets for validation study, but there are technical challenges to this approach [38].

In summary, we identified certain urinary lncRNA targets that may help the identification of lupus nephritis and predict the histological class of nephritis. Our findings indicate that urinary lncRNA levels may be developed as biomarkers for lupus nephritis.

## Figures and Tables

**Figure 1 ijms-24-11813-f001:**
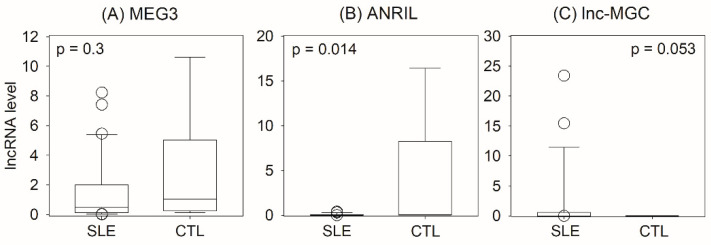
Comparison of urinary lncRNA levels between patients with systemic lupus erythematosus (SLE) and healthy controls (CTL): (**A**) maternally expressed gene 3 (MEG3); (**B**) antisense non-coding RNA in the INK4 locus (ANRIL); (**C**) long non-coding megacluster (lnc-MGC). Data were compared using the Mann–Whitney U test.

**Figure 2 ijms-24-11813-f002:**
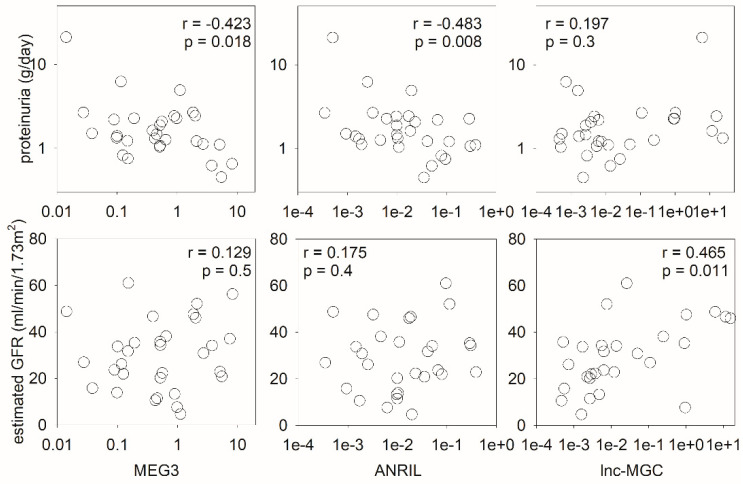
Scatter plots depicting the correlation matrix between urinary lncRNA levels (MEG3, ANRIL, and lnc-MGC) and clinical parameters (proteinuria and estimated GFR). Data were analyzed using the Spearman’s rank correlation coefficient (GFR, glomerular filtration rate).

**Figure 3 ijms-24-11813-f003:**
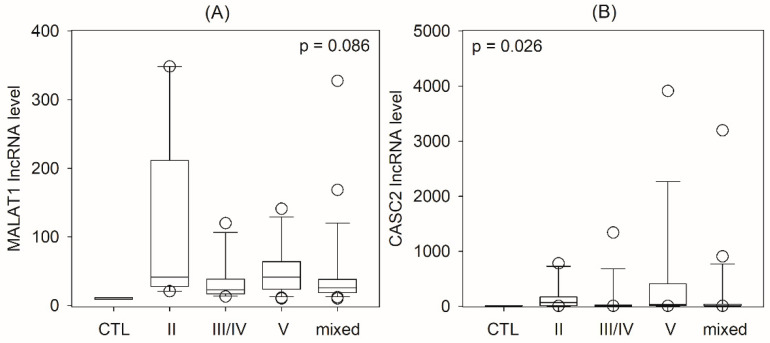
Comparison of urinary lncRNA levels between patients with different classes of lupus nephritis: (**A**) metastasis-associated lung carcinoma transcript 1 (MALAT1); (**B**) cancer susceptibility candidate 2 (CASC2). Data were compared using the Kruskal–Wallis test (CTL, healthy control group; mixed indicates the coexistence of proliferative and membranous nephritis).

**Figure 4 ijms-24-11813-f004:**
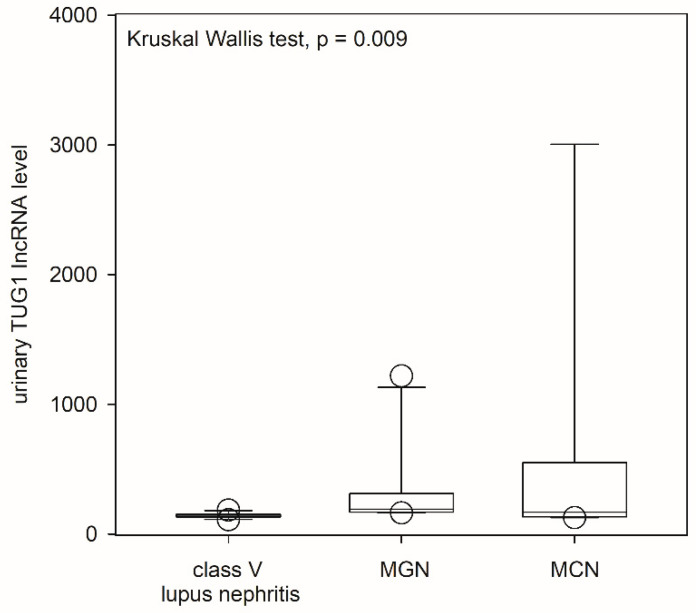
Comparison of urinary lncRNA taurine-upregulated gene 1 (TUG1) levels between patients with different causes of nephrotic syndrome. Data were compared using the Kruskal–Wallis test (MGN, membranous glomerulonephritis; MCN, minimal change nephropathy).

**Figure 5 ijms-24-11813-f005:**
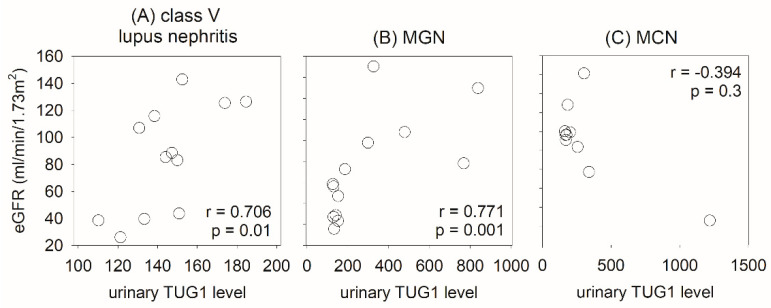
Scatter plots depicting the correlation matrix between urinary lncRNA taurine-upregulated gene 1 (TUG1) levels and estimated glomerular filtration rate (eGFR) in patients with different causes of nephrotic syndrome: (**A**) class V lupus nephritis; (**B**) membranous glomerulonephritis (MGN); (**C**) minimal change nephropathy (MCN). Data were analyzed using the Spearman’s rank correlation coefficient.

**Table 1 ijms-24-11813-t001:** Baseline characteristics of the patients.

	SLE	Control
No. of cases	31	6
Age	45.6 ± 12.3	43.5 ± 10.1
Sex (F:M)	29:2	3:3
eGFR (mL/min/1.73 m^2^)	30.8 (20.2–38.1)	93.8 (69.1–113.3)
Proteinuria (g/day)	1.4 (1.1–2.3)	--
Glomerulosclerosis (%)	1.5 (0.0–30.0)	
Tubulointerstitial fibrosis (%)	10.0 (5.0–15.0)	
SLEDAI	12 (9–14)	
Activity index	2 (0–8)	
Chronicity index	2 (1–3)	

eGFR, estimated glomerular filtration rate; SLEDAI, systemic lupus erythematosus disease activity index.

**Table 2 ijms-24-11813-t002:** Baseline characteristics of patients in cohort #2.

	SLE					
	All Case	Class II	Class III or IV	Class V	Mixed	Control
No. of cases	78	10	15	29	24	7
Age	41.7 ± 12.6	37.3 ± 14.9	40.7 ± 11.8	44.9 ± 11.7	40.3 ± 13.1	44.4 ± 8.3
Sex (F:M)	73:5	9:1	14:1	29:0	21:3	6:1
eGFR (mL/min/1.73 m^2^)	47.4 (33.0–67.9)	55.1 (26.1–85.7)	44.1 (31.1–64.0)	55.4 (36.8–72.5)	45.8 (29.6–73.9)	93.4 (92.4–104.5)
Proteinuria (g/day)	1.9 (1.2–3.4)	1.2 (0.9–2.1)	2.1 (1.4–5.1)	2.1 (1.3–4.8)	1.5 (1.2–2.2)	--
Glomerulosclerosis (%)	0.6 (0.0–3.4)	0.0 (0.0–8.3)	2.7 (0.4–5.2)	0.0 (0.0–2.2)	1.5 (0.0–3.3)	
Tubulointerstitial fibrosis (%)	1.0 (0.0–6.6)	0.0 (0.0–16.2)	4.0 (1.2–7.1)	0.0 (0.0–4.1)	1.3 (0.0–7.9)	
SLEDAI	11 (9–13)	10 (8–11)	11 (9–13)	12 (9–14)	11 (9–14)	
Activity index	3 (1–7)	1 (0–2)	7 (6–10)	1 (1–3)	7 (4–9)	
Chronicity index	1 (0–2)	0 (0–3)	1 (1–2)	0 (0–1)	1 (0–2)	

eGFR, estimated glomerular filtration rate; SLEDAI, systemic lupus erythematosus disease activity index.

**Table 3 ijms-24-11813-t003:** Baseline characteristics of the patients in cohort #3.

	SLE	MGN	MCN
No. of cases	12	14	10
Age	59.7 ± 16.1	60.5 ± 14.1	59.1 ± 16.1
Sex (F:M)	9:3	5:9	6:4
eGFR (mL/min/1.73 m^2^)	87.0 (40.6–123.0)	65.5 (28.4–118.4)	116.4 (97.1–127.3)
Proteinuria (g/day)	5.8 (1.3–10.0)	1.5 (0.8–4.0)	6.3 (4.9–10.5)
Glomerulosclerosis (%)	0 (0–12)	0 (0–12)	0 (0–9)
Tubulointerstitial fibrosis (%)	0 (0–10)	2 (0–12)	0 (0–2)
SLEDAI	7 (5–10)	--	--
Activity index	0 (0–2)	--	--
Chronicity index	0 (0–1)	--	--

eGFR, estimated glomerular filtration rate; SLEDAI, systemic lupus erythematosus disease activity index.

## Data Availability

All data generated or analyzed during this study are included in this article. Further enquiries can be directed to the corresponding author.

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
