# Peer review of "Urinary Long Non-Coding RNA Levels as Biomarkers of Lupus Nephritis"

_ijms, 2023, doi:10.3390/ijms241411813_

Round 1

Reviewer 1 Report

The authors report novel biomarkers of lupus nephritis using the molecular diagnostic test, real-time PCR. However, there are some concerns regarding the scientific significance and its presentation.

1. Introduction part require more details particularly what is the functional significant of each selected lncRNA.

2. The manuscript describe different expression of long non-coding RNA as the potential biomarker. However, there was no receiver-operator-curve (ROC) which is essentially a good tool for biomarker discovery.

3. What is the normal range of lncRNA ? why was the mRNA level widely range from single-digit number (0-10 for MEG3) up to 4 digit number (for CASC2). Should we have a cut-off levels for normal expression level.

4. In this work, the range of eGFR for nephritis was very unusual in figure 2. In general, active lupus nephritis should have low eGFR (less than 80 mL/min/1.73m2)

5. The authors explained TUG1 may involved in the glomerular barrier damage. However, it is interesting why urine TUG was associated with eGFR. What is the possible mechanisms.

6. In the discussion, the authors did not mention where was these lnc RNA come from.  It is interesting to know which urinary cell type expresses lnc RNA.

Author Response

Reviewer #1

  1. Introduction part require more details particularly what is the functional significant of each selected lncRNA.

Page 3, last 5 lines; Page 4, line 1-2: As suggested, we elaborate on the functional significance of each lncRNA target in the background.

  1. The manuscript describe different expression of long non-coding RNA as the potential biomarker. However, there was no receiver-operator-curve (ROC) which is essentially a good tool for biomarker discovery.

Page 9, paragraph 1, last 2 lines; Page 10, paragraph 1, last 5 lines; Page 10, paragraph 2, line 4-7: As suggested, we add the ROC curve analysis and present the possible cut-off value whenever possible.

  1. What is the normal range of lncRNA ? why was the mRNA level widely range from single-digit number (0-10 for MEG3) up to 4 digit number (for CASC2). Should we have a cut-off levels for normal expression level.

Page 14, paragraph 1, line 2-4: The area of urinary lncRNA is new and there is no normal range reported. Our result indicates that the levels vary from almost undetectable to a substantial amount according to the specific lncRNA target. This point is highlighted in the discussion.

  1. In this work, the range of eGFR for nephritis was very unusual in figure 2. In general, active lupus nephritis should have low eGFR (less than 80 mL/min/1.73m2)

Figure 2: We are deeply sorry that there was a mistake we used in the formula for the calculation of eGFR in this part. The data is corrected. The result of analysis was not affected.

  1. The authors explained TUG1 may involved in the glomerular barrier damage. However, it is interesting why urine TUG was associated with eGFR. What is the possible mechanisms.

Page 13, last 3 lines: As suggested, we postulate the possible reason of association between TUG1 and eGFR.

  1. In the discussion, the authors did not mention where was these lnc RNA come from. It is interesting to know which urinary cell type expresses lnc RNA.

Page 14, paragraph 1, line 10-14: We did not determine the cellular origin of the lncRNA in the urine. Based on our previous study, infiltrating mononuclear cells and renal tubular cells are the major origin of mRNA. This point is elaborated in the discussion.

Reviewer 2 Report

The authors have addressed a very important and interesting problem towards the SLE which is responsible for the substantial morbidity and mortality. Early diagnosis using the non-invasive biomarkers could be great in case of lupus nephritis. However, I have the following comments which can help to improve the overall quality of the study.

1. Consider rewriting the abstract by explaining the cohorts information in terms of number of patients.

2. The flow of the abstract can be improved.

3. Introduce some information about the preprocessing of the matrices in the abstract.

4. The background in the manuscript can be improved by explaining the overall workflow in the comprehensive way. The authors have mentioned only one line regarding their approach.

5. Please explain how patients with other glomerular diseases act as control for this study.

6. Since data comes from three different cohorts, there could be chance of batch effect. Did authors removed or rectify the batch effect before performing the downstream analysis.

7. Is there a particular reason to use the non-parametric test such as Kruskal-Wallis, and Mann Whitney U tests. Have authors run the normality test before deciding to apply the non-parametric tests.

8. Please explain the rationale behind using the spearman's rank-order correlation, as the values are contnious in nature. Why authors have not used the Pearson's correlation test.

9 Please explain the significance of lncRNA targets MEG3, ANRIL, and MGC for the identification of lupus nephritis.

10. In the legend of Figure 1, change "Date" to "Data".

11. Please use the Pearson correlation instead of spearman to check if the hypothesis still holds.

12. The authors can apply the post-hoc test to apply the multiple comparison to find the actually significant histological class.

13. Please provide the rational behind the positive trend of correlation between TUG1 and eGFR in case of Class V lupus nephritis and MGN, but negative in case of MCN.

14. Please provide the conclusion of the study in a separate section.

15. Please explain the limitation of the study as a differnt section.

1. There are many punctuation errors throughout the manuscript. Please rectify them.

2. I would advise authors to rectify the grammatical errors.

Author Response

Reviewer #2

  1. Consider rewriting the abstract by explaining the cohorts information in terms of number of patients.

Page 2: The abstract is extensively re-written and provide the number of patients in each part of the study.

  1. The flow of the abstract can be improved.

Page 2: The abstract is extensively re-written to improve the flow of presentation.

  1. Introduce some information about the preprocessing of the matrices in the abstract.

Page 2: The abstract is extensively re-written to improve the flow of presentation.

  1. The background in the manuscript can be improved by explaining the overall workflow in the comprehensive way. The authors have mentioned only one line regarding their approach.

Page 4, line 3-6: As suggested, we elaborate on the overall idea and workflow of the study in the background.

  1. Please explain how patients with other glomerular diseases act as control for this study.

Page 5, paragraph 1, line 6-7: As suggested, we describe in more details about the use of other glomerular disease as controls.

  1. Since data comes from three different cohorts, there could be chance of batch effect. Did authors removed or rectify the batch effect before performing the downstream analysis.

Page 14, paragraph 1, line 14-16: We agree that since data of our study come from three different cohorts, there could be chance of batch effect. Nonetheless, we did not attempt to remove or rectify the batch effect but treat them as three separate studies as there was no overlap in the lncRNA targets tested between the cohorts. This point is elaborated in the discussion.

  1. Is there a particular reason to use the non-parametric test such as Kruskal-Wallis, and Mann Whitney U tests. Have authors run the normality test before deciding to apply the non-parametric tests.

Page 7, last 5 lines: We have confirmed that the distribution of all lncRNA levels and many clinical parameters (eGFR and proteinuria) were not in normal distribution, and non-parametric tests, including the use of Spearman’s correlation coefficient, were recommended by our statistician.

  1. Please explain the rationale behind using the spearman's rank-order correlation, as the values are contnious in nature. Why authors have not used the Pearson's correlation test.

Page 7, last 5 lines: We have confirmed that the distribution of all lncRNA levels and many clinical parameters (eGFR and proteinuria) were not in normal distribution, non-parametric tests, including the use of Spearman’s correlation coefficient, were recommended by our statistician.

9 Please explain the significance of lncRNA targets MEG3, ANRIL, and MGC for the identification of lupus nephritis.

Page 13, paragraph 1, line 3-5: As suggested, we explain the significance (actually the lack of clinical significance) of the lncRNA targets MEG3, ANRIL, and MGC.

  1. In the legend of Figure 1, change "Date" to "Data".

The typo is rectified. We are sorry for the mistake.

  1. Please use the Pearson correlation instead of spearman to check if the hypothesis still holds.

We tried analysis with Pearson’s correlation and the result was similar. Nonetheless, since the distribution of all lncRNA levels and many clinical parameters (eGFR and proteinuria) were not in normal distribution, Spearman’s correlation coefficient was recommended by our statistician.

  1. The authors can apply the post-hoc test to apply the multiple comparison to find the actually significant histological class.

Page 10, paragraph 2, line 3-5: Post hoc analysis and adjustment for multiple comparison actually did not show any significant difference between subgroups. This point is clarified in the result.

  1. Please provide the rational behind the positive trend of correlation between TUG1 and eGFR in case of Class V lupus nephritis and MGN, but negative in case of MCN.

Page 13, last 3 lines: As suggested, we postulate the possible reason of association between TUG1 and eGFR. In short, since TUG1 may be relevant for immune complex deposit, it seems reasonable to see an effect on lupus nephritis and MGN, but not MCN.

  1. Please provide the conclusion of the study in a separate section.

Page 14, last paragraph: As suggested, we provide the conclusion in a separate section.

  1. Please explain the limitation of the study as a differnt section.

Page 14, paragraph 1: As suggested, we discuss the limitations in a separate section.

  1. There are many punctuation errors throughout the manuscript. Please rectify them.

As suggested, the punctuations are checked and corrected by an independent expert.

  1. I would advise authors to rectify the grammatical errors.

As suggested, the grammar is checked and corrected by an independent expert.

Reviewer 3 Report

In this article, the authors tried to use long non-coding  RNA (lncRNA) as non invasive biomarker for lupus nephritis. They divided their study into three studies. In the first study, they tested the urinary levels of lncRNA targets MEG3, ANRIL, and lnc-MGC in 31 lupus patients and 6 controls. In the second study, they tested lncRNA targets MALAT and CASC2 in 78 lupus nephritis patients with various histological classes, and 7 healthy controls. In the third study, urinary level of lncRNA TUG1 was compared between 36 patients with pure class V lupus nephritis, primary membranous nephropathy, or minimal change nephropathy.

Although the authors discussed the limitations of the study in the discussion part but,

1- The introduction part need more details about the lncRNA targets used.

2- Materials and methods stated that kidney's staining has been done. It will be good to show kidney's data 

3- It will be beneficial to add all the data that not shown as a supplementary

4- Did the authors have blood samples from these patients? or have data of their autoantibody levels?

5- Did the authors tried to measure cytokines levels in the urine?

Author Response

Reviewer #3

1- The introduction part need more details about the lncRNA targets used.

Page 3, last 5 lines; Page 4, line 1-2: As suggested, we elaborate on the functional significance of each lncRNA target in the background.

2- Materials and methods stated that kidney's staining has been done. It will be good to show kidney's data

The Jones’ silver stain was only used for the quantification of kidney fibrosis but not intra-renal lncRNA levels. We are sorry for the confusion.

3- It will be beneficial to add all the data that not shown as a supplementary

Supplementary Table 1: As suggested, we include all data in a supplementary table.

4- Did the authors have blood samples from these patients? or have data of their autoantibody levels?

Page 14, paragraph 1, line 10-11: Unfortunately we do not have blood samples for the measurement of lncRNA levels. This point is elaborated in the discussion.

5- Did the authors tried to measure cytokines levels in the urine?

Page 14, paragraph 1, line 10-11: Unfortunately we did not measure urinary or serum cytokine level. This point is elaborated in the discussion.

Round 2

Reviewer 2 Report

The authors have replied to my questions satisfactorily. I am happy to accept the manuscript in the current form.

Reviewer 3 Report

The authors modified the manuscript to address all the required points. Thanks